# A Simple Red Tide Monitoring Method using Sentinel-2 Data for Sustainable Management of Brackish Lake Koyama-ike, Japan

**Yuji Sakuno** [1,*], **Akihiro Maeda** [2], **Akihiro Mori** [3], **Shuji Ono** [4] and **Akihiro Ito** [5]

1   Graduate School of Engineering, Hiroshima University, 1-4-1, Kagamiyama, Higashi-Hiroshima, Hiroshima 739-8527, Japan

2   Tottori Prefectural Institute of Public Health and Environmental Science, 526-1, Minamidani, Yurihama-cho, Tottori 682-0704, Japan; maedaa@pref.tottori.lg.jp

3   Tottori Prefecture Water Environment Management Division, 1-220, Higashimachi, Tottori, Tottori 680-8570, Japan; moriak@pref.tottori.lg.jp

4   Imaging Technology Center, FUJIFILM Corporation, 798, Miyanodai, Kaisei-machi, Ashigarakami-gun, Kanagawa 258-8538, Japan; shuji.ono@fujifilm.com

5   Corporate Sales and Marketing Division, NTT DOCOMO, INC., 1-8-1, Akasaka, Minatoku, Tokyo 107-0052, Japan; akihiro.itou.cf@s1.nttdocomo.com

*   Correspondence: sakuno@hiroshima-u.ac.jp; Tel.: +81-82-424-7773

**Abstract:** We proposed and validated a method for monitoring red tides in the brackish Lake Koyama-ike, Japan, using Sentinel-2 Multispectral Instrument (MSI) data with a 10 m spatial resolution. To achieve this objective, we acquired 36 spectral reflectance/Chla data points in the field from 2012 to 2018. We obtained a high correlation of Chla ($R^2 = 0.83$) using the proposed red tide model ($RI_{KY}$ = [MSI Band 5 − MSI Band 4]/[MSI Band 5 + MSI Band 4]) and field data. Based on our results, the proposed model was also validated using five Sentinel-2/Chla datasets from April to August 2017. Chla and red tide distribution characteristics estimated from Sentinel-2 data hardly appeared from April to July, and then spread rapidly throughout the lake (more than 70%) in August. Thus, Sentinel-2 data proved to be a very powerful tool in monitoring red tides in Lake Koyama-ike.

**Keywords:** red tide; chlorophyll; Sentinel-2; brackish lake; atmospheric correction

## 1. Introduction

Lake Koyama-ike, on the outskirts of Tottori, Japan, was changed from fresh water (about 0.2 psu) to brackish water (about 4 to 9 psu) when the entire seawater tide gate was opened on 12 March 2012. The odor and landscape deterioration due to cyanobacteria (phytoplankton forming blue-green algae) that generated the summer before opening the gate was eliminated. However red tides consisting mostly of diatoms and dinoflagellates were generated after opening the gate [1]. Red tide countermeasures thus became a crucial theme for the administration, as clam (*Corbiculidae* spp.) fisheries became more active. There was a need for a visualization system that could convey red tide distributions to fishers and managers in a quick and easy-to-understand manner. However, spatial and temporal water quality variations in the brackish water lake are extremely intense, making it difficult to grasp the entire area by ship-based field surveys. Therefore, understanding the environment through remote sensing is necessary. Thus far, researchers have attempted to develop a remote sensing chlorophyll-a (Chla) estimation algorithm using spectral reflectance surveys and drones, aquatic plant mapping, and other methods in the lake and its surroundings [2,3]; however, these efforts have not yet yielded a satellite-based monitoring system to estimate red tide distribution.

High spatial resolution is necessary for Chla estimation by satellite remote sensing in relatively small lakes (<5 km$^2$ square). Chla estimation has been actively tested using Landsat 8 [4,5] and the Advanced Spaceborne Thermal Emission and Reflection Radiometer (ASTER) NASA's Terra satellite [6,7], which have spatial resolutions of 15–30 m. However, these satellites do not have adequate observation repeat cycles (16-day cycle) and observation wavelength (no observations near 700 nm, which is the wavelength advantageous for coastal Chla estimation). In contrast, the European Space Agency's (ESA) Sentinel-2 satellites [8] has a 10 m spatial resolution and 10 day temporal resolution (every five days with two satellites). The two satellites of the Sentinel-2 constellation were launched in June 2015 and March 2017. Additionally, because Sentinel-2 has a 705 nm observation wavelength, which is advantageous for Chla estimation in turbid water, there is a high possibility of relatively short-period red tide monitoring, even in small brackish lakes. Chla estimation research using Sentinel-2 was recently conducted in waters around the world [9,10] but because it is new, its case study is extremely significant compared to research using Landsat data as there has been little research on red tide monitoring in brackish lakes. Additionally, it is necessary to consider atmospheric correction methods to process the red tide situation quickly and simply (or automatically) with a 10 m spatial resolution every 10 days (every five days with two satellites).

Based on this background, we propose and validate a red tide monitoring method for sustainable management of Lake Koyama-ike using high-resolution Sentinel-2 data. Specifically, we propose a red tide detection model based on Sentinel-2 data and spectral reflectance results measured locally during without red tide and with red tide, and to validate it using other field data. We also present a simple method to perform atmospheric corrections in the study area that generally require advanced calculations, and validated this atmospheric correction method using actual Sentinel-2 data.

## 2. Materials and Methods

### 2.1. Study Area

The Lake Koyama-ike is in the coastal area of Tottori Plain, as shown in Figure 1. It has an area of 7.0 km$^2$, an average water depth of 2.8 m, a maximum water depth of 6.5 m, and a lake shore extension of 8 km. Relatively large islands, such as Ao-shima Island (gourd-type island in the center of the lake), Tsubu-shima Island, and Dango-shima Island are scattered throughout the lake. There are 6 inflow rivers, such as the Fukui, Nagara, and Miyamaguchi rivers, all of which flow from mountains to the south or west. Among these, the Nagara River basin on the lake's western part is the largest. Lake Koyama-ike was connected to the Sea of Japan through the Sendai River until 1983, but was directly connected to the Sea of Japan after construction at the mouth of the Sendai River. Because the lake was directly connected to the sea, surrounding fields experienced salt damage. Thus, the tide gate was responsible for salt adjustment (downstream of the Koyama River described in Figure 1), and until the gate was open the lake water was mostly fresh. The gate was opened for to delate for odors and improve fisheries. Before the tide gate was opened, a large amount of blue-green algae and aquatic plants would grow from spring to autumn, causing bad odors and adversely affecting the fishery. However, opening the gate has brought problems such as red tides and poor oxygenation in the bottom of the lake because of the rapid increase in salinity (up to about 1/3 of seawater). The red tides in Lake Koyoma-ike were often attributed to the dominance of the dinoflagellate *Heterocapsa* (Mori et al. Unpublished) and other red tide-forming species such as *Cheatoceros*, *Cyclotella*, and *Skeletonema*. Dominance of many of these phytoplankton has been confirmed [1]. Furthermore, a new dinoflagellate species, *Alexandrium ostenfeldii*, was discovered in 2013 [11].

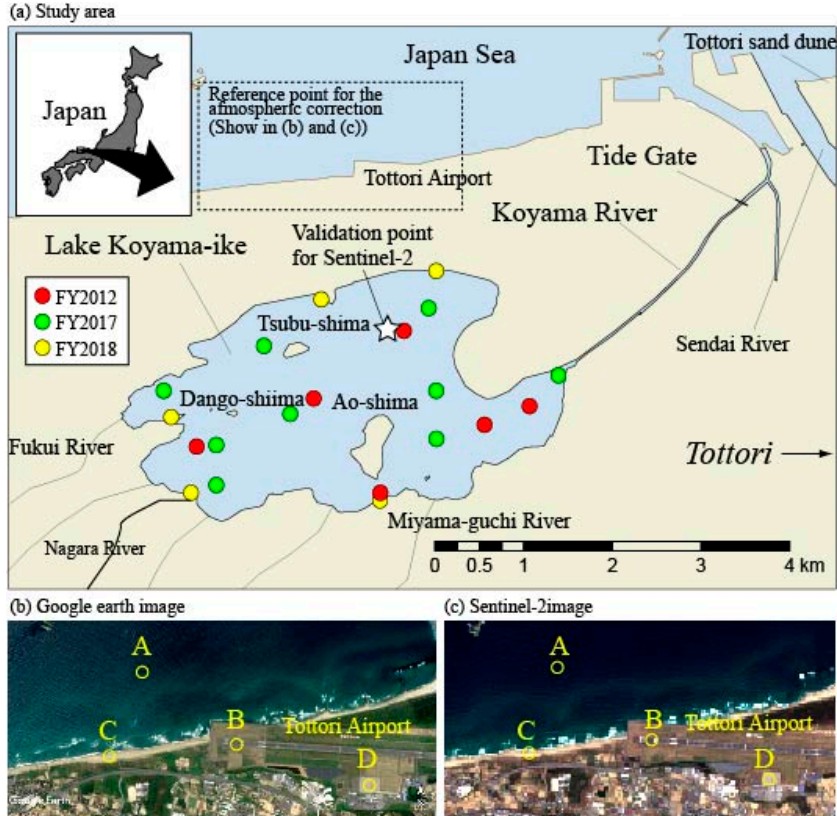

**Figure 1.** Study area in Lake Koyama-ike, Japan with field observation points (**a**), reference points for relative atmospheric correction layered on a Google Earth image (**b**), and reference points layered on a Sentinel-2 image (**c**).

*2.2. Field Survey*

The field campaigns were conducted on the 19 July and 25 September 2012 (without red tide conditions), and on the 4 September 2017 and 22 August 2018 (with red tide conditions). There were 6 to 15 measurements of spectral reflectance/Chla and suspended substance (SS), which were collected during each campaign as shown in Figure 1. We used a MS 720 portable spectral radiometer (manufactured by EKO, observed wavelength range: 350–1050 nm, wavelength interval: 3.3 nm, half width: 10 nm, Tokyo, Japan) and a reference white board (Labsphere, ~ 13 cm) to measure spectral reflectance of the surface water. We measured spectral irradiance from the water surface and the reference white board from the ship 3 times with a spectrometer aperture angle of 90° and a measurement distance of approximately 15 cm. Analysis software (SpectroManager software ver. 1.4, EKO, Tokyo, Japan) attached to the spectrometer converted irradiance data measured every 3.3 nm to data every 1 nm. Remote sensing reflectance (Rrs) was calculated as the average of 3 ratios of irradiances from the water surface and the reference white board divided by the circle constant ($\pi$). Figure 2 shows the spectral reflectance of all measured points. On the other hand, we measured SS and Chla in the laboratory using a 1 liter surface water sample collected from the ship. We measured SS using the JIS (Japanese Industrial Standard) method that utilizes differences in the weight of filter paper before and after filtering sample water. Additionally, we measured Chla using acetone extraction and absorption spectrophotometry and quantified it by calculating Chla based on the SCOR/UNESCO standard. Table 1 showed the ranges of acquired Chla and SS data. In this study we defined red tide as Chla exceeding 20 µg/L, which was the boundary between the day when a red tide was confirmed and the day when it was not confirmed. Thus, it was judged that on 4 September 2017 and 22 August 2018, all stations and 4 out of 6 stations had red tide conditions, respectively. Additionally, during the red tide in 2017, we boarded a disaster prevention helicopter on September 4 2017 to take photographs of

the red tide from the air, and then collected surface water samples, which contained dinoflagellate *Scrispsiella rotunda* (the number of cells at the same point is $5.3 \times 10^3$ cells mL$^{-1}$), a red tide plankton species. The results of this survey were reported in detail by Sakuno et al. [12].

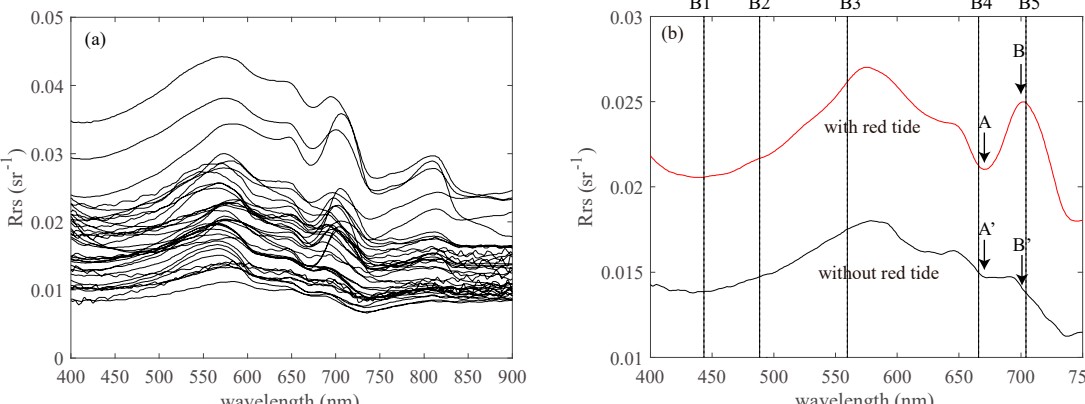

**Figure 2.** Field spectral reflectance characteristic in Lake Koyama-ike, Japan from 2012 to 2018. (**a**) All data, (**b**) comparison of average spectral reflectance for without red tide conditions using nine data points on 25 September 2012 and with red tide conditions using six data points on 4 September 2017.

**Table 1.** Chla and suspended substance (SS) ranges of field data.

| No. | Date | Chla (μg/L) | | SS (mg/L) | | N |
|-----|------|-----|-----|-----|-----|---|
| | | **Min** | **Max** | **Min** | **Max** | |
| 1 | 19 July 2012 | 4.7 | 15.1 | 3.6 | 47.6 | 15 |
| 2 | 25 September 2012 | 7.8 | 18.8 | 4.0 | 29.6 | 9 |
| 3 * | 4 September 2017 | 37 | 204 | 9.3 | 22.8 | 6 |
| 4 * | 22 August 2018 | 6.8 | 61.6 | 7.8 | 38.3 | 6 |
| | Total | 4.7 | 204 | 3.6 | 48 | 36 |

\* red tide bloom condition

### 2.3. Red tide Bloom Algorithms

Recent red tide detection methods using satellite ocean color sensors represented by the MODerate resolution Imaging Spectroradiometer (MODIS), Sea-Viewing Wide Field-of-View Sensor (SeaWiFS), and MEdium Resolution Imaging Spectrometer (MERIS) instruments were reviewed in detail by Blondeau-Patissier et al. [13]. That paper detailed many studies for estimating Chla, which is an index of red tide, using either the reflectance ratio or differences between 2 or 3 wavelengths. For example, the blue-green reflectance ratio used by SeaWiFS [13] and the red-to-infrared reflectance ratio used by MERIS [14] were known as typical Chla estimation models. The latter of these was especially effective as a Chla estimation model in coastal waters rich in SS and colored dissolved organic matter (CDOM). For example, when using Rrs of the Sentinel-2 observation wavelength, Chla can be expressed as

$$Chla \propto \frac{R_{rs}(705)}{R_{rs}(665)} \tag{1}$$

These models have a Chla estimation accuracy of approximately 35% or less. On the other hand, Ahn and Shanmugam [15] proposed a red tide estimation model (RI: Tide index) using the SeaWiFS band as an example in northeast Asian waters. This model was indexed by normalizing the 510 nm and 550 nm reflectance ratios with strong absorption and no change at 443 nm in the red tide.

$$RI_{Ahn} = \frac{[L_w(510)/L_w(555) - L_w(443)]}{[L_w(510)/L_w(555) + L_w(443)]} \tag{2}$$

When Sentinel-2 Rrs data was input into Equation (2) can be represented by the following model (referred to as Model 1).

$$RI_{Model1} = \frac{[R_{rs}(490)/R_{rs}(560) - L_w(443)]}{[R_{rs}(490)/R_{rs}(560) + L_w(443)]} \quad (3)$$

On the other hand, another study [16] showed features of the red tide spectrum [nLw (443)/nLw (490)] and [nLw (490)/nLw (555)] (nLw: Normalized water-leaving radiances) in the Seto Inland Sea of Japan and devised a method for distinguishing red tides from non-red tides. Transforming this model into the form of Model 1 yields

$$RI_{Takahashi} = \frac{[nL_w(443)/nL_w(490) - nL_w(490)/nL_w(555)]}{[nL_w(443)/nL_w(490) + nL_w(490)/nL_w(555))]} \quad (4)$$

Furthermore, inputting Sentinel-2 Rrs data into Equation (4) yields the following model (referred to as Model 2).

$$RI_{Model2} = \frac{[R_{rs}(443)/R_{rs}(490) - R_{rs}(490)/R_{rs}(560)]}{[R_{rs}(443)/R_{rs}(490) + R_{rs}(490)/R_{rs}(560)]} \quad (5)$$

In previous work, we demonstrated the effectiveness of the red-near infrared model, as shown in Equation (1), in Lake Koyama-ike [2]. Therefore, we proposed the following model (referred to as [$RI_{KY}$]) to give it the same form as Models 1 and 2.

$$RI_{KY} = \frac{[R_{rs}(705) - R_{rs}(665)]}{[R_{rs}(705) + R_{rs}(665)]} \quad (6)$$

### 2.4. Sentinel-2/Chla Data Set

For this research, we downloaded level 1C images of 5 scenes shown in Table 2 from April to August 2017 from Earth Explorer operated by the U.S. Geological Survey. The spatial resolution of the bands of MultiSpectral Instrument (MSI) radiance information shown in Table 3 in was 10–60 m. The level 1C product used a UTM WGS 84 projection consisting of 100 km x 100 km angle-corrected tiles, with images for each band stored as separate JPEG 2000 files. The spatial resolution of Band 4 has been scaled down to 20 m using average method to unify Band 4 and Band 5 resolutions used for Chla calculations.

**Table 2.** Sentinel-2/Chla data set in 2017.

| No. | Date | Observation Time (JST) | Mean Sun Angle Zenith/Azimuth | Chla (µg/L) |
|-----|------|------------------------|-------------------------------|-------------|
| 1 | 4 April | 10:47 | 33.7°/148.6° | 6.7 |
| 2 | 24 April | 10:47 | 26.8°/144.3° | 7.2 |
| 3 | 14 May | 10:47 | 21.7°/137.7° | 3.3 |
| 4 | 13 July | 10:47 | 20.6°/127.4° | 21.5 |
| 5 | 2 August | 10:47 | 23.8°/134.2° | 34.3 |

Sentinel-2's amosperic correction was often processed using ESA's Sen2Cor algorithm [17] with dedicated software such as the Sentinel Application Platform, which is a free Sentinel-2 data processing tool. However, data processing was complicated, and aerosol correction suited to the local environment were not necessarily performed. In contrast, simple atmospheric correction without the use of special tools, such as atmospheric correction with Dark Object Subtraction (DOS) [18,19] and relative atmospheric correction [20–23], has been widely used. Among these, relative atmospheric correction was considered to be more accurate because correction was performed with pixel values of a plurality of points whereas correction was performed with one dark pixel value in DOS.

**Table 3.** Sentinel-2 multispectral instrument performance.

| Band | Wavelength (nm) | Spatial Resolution (m) | Band | Wavelength (nm) | Spatial Resolution (m) |
|------|-----------------|------------------------|------|-----------------|------------------------|
| B1 | 443 | 60 | B 6 | 740 | 20 |
| B2 | 490 | 10 | B 7 | 775 | 20 |
| B3 | 560 | 10 | B 8 | 842 | 20 |
| B4 | 665 | 10 | B 8a | 865 | 10 |
| B5 | 705 | 20 | | | |

Relative atmospheric correction finds the relative relationship between data points belonging to several types of ground coverings with low reflectance, such as open ocean, and high reflectance points without changes in reflection characteristics over time. Relative atmospheric correction is a method for correcting the relative reflectance relationships between ground reference points, which are ground covering objects with reflection characteristics that are assumed not to change temporally. In this research, we used data from 2 August 2017, where large-scale red tides were expected as a reference and determined the conversion coefficient of each data point for each band by regression calculation using sensor output value (digital count [DC] less than that of the reference object). The relative atmospheric correction was originally desirable based on radiance or reflectance as described in references [22,23]. However, since we aimed at a simple method in this research, it was calculated by a DN (Digital Number)-based method [20,21]. We used 4 points (see Figure 1b,c) commonly used in relative atmospheric correction as ground reference objects: (a) Open ocean (Japan Sea), (b) asphalt (Tottori airport), (c) sand (coast), and (d) concrete (Tottori airport). Additionally, we used the surface Chla value automatically observed by Tottori Prefecture at 134.15732° E, 35.5172° N (star in Figure 1) as local Chla data for validation of Sentinel-2 Chla. The surface Chla data were measured using a Hydrolab 5DS manufactured by OTT Hydromet GmbH, which has a measurement range of 0.03–500 µg/L, a measurement accuracy of ± 3%, and a resolution of 0.01 µg/L.

## 3. Results

### 3.1. Validation of the Red Tide Model Using Field Data

First, we confirmed the positional relationship between spectral reflectance characteristics in the lake and MSI's observation wavelength. Figure 2 shows the total spectral reflectance obtained in th study area. Figure 2a shows the total spectral reflectance data and Figure 2b shows the average value for non-red tide (25 September 2017) and red tide (4 September 2017) dates. From this, we obtained spectral characteristics for the lake with a maximum around 560–570 nm and local minima and maxima of characteristic reflectance can be confirmed, respectively, at 660–670 nm (A) and 700–710 nm (B). Peak wavelengths A and B show good agreement with MSI band 4 and band 5.

Next, we investigated whether the wavelengths of the MSI band 4 and band 5 for $RI_{KY}$ in Equation (6) are appropriate. Figure 3 shows a matrix of correlation coefficients ($R^2$) between Chla and the following $RI_{xy}$ when combining the two bands at 1 nm intervals in the 400–900 nm range to investigate suitability.

$$RI_{xy} = \frac{[R_{rs}(\text{Band } 2) - R_{rs}(\text{Band } 1)]}{[R_{rs}(\text{Band } 2) + R_{rs}(\text{Band } 1)]} \tag{7}$$

Figure 3a shows $R^2$ with respect to Chla and Figure 3b shows $R^2$ with respect to SS. From this we see that Chla showed high correlation ($R^2 > 0.6$) when the two bands combined in the 600 to 750 nm range area, which is indicated by the dotted square. In the area of high correlation, only the combination of band 4 and band 5 applies when only the MSI band is considered. In contrast, SS was uncorrelated for wavelength combination.

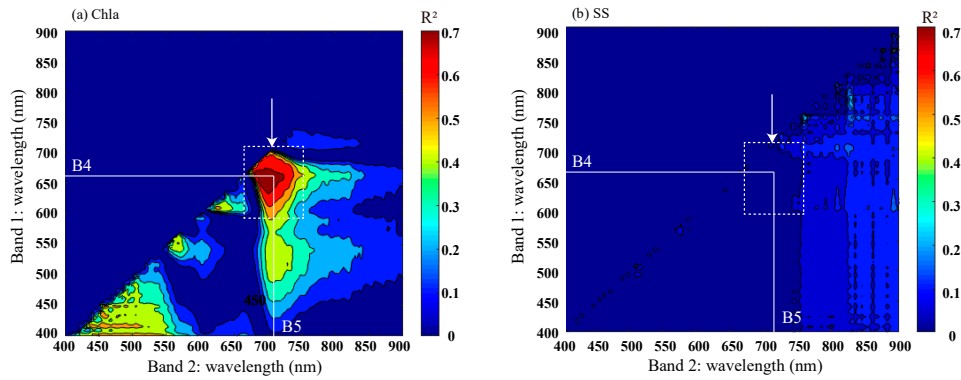

**Figure 3.** Correlation coefficient ($R^2$) matrix between tide index (*RI*) model using Equation (6) combining two bands at 1 nm and water quality parameter. (**a**) Chlorophyll A, (**b**) suspended substances.

Figure 4a shows the relationship between $RI_{KY}$ and Chla for MSI band 4 and band 5 validated by the results shown in Figure 3. For comparison, the relationships between this model and SS, this model and $RI_{model1}$, and this model and $RI_{model2}$ are shown in Figure 4b–d, respectively. From this, we see that $RI_{KY}$ obtained a higher correlation ($R^2 = 0.85$) in the exponential regression model than the linear regression model ($R^2 = 0.73$). In contrast, SS was uncorrelated ($R^2 = 0.00$) even in the exponential regression model. Furthermore, we did not obtain high correlations ($R^2 = 0.00$ and $R^2 = 0.25$) in models based on previous studies.

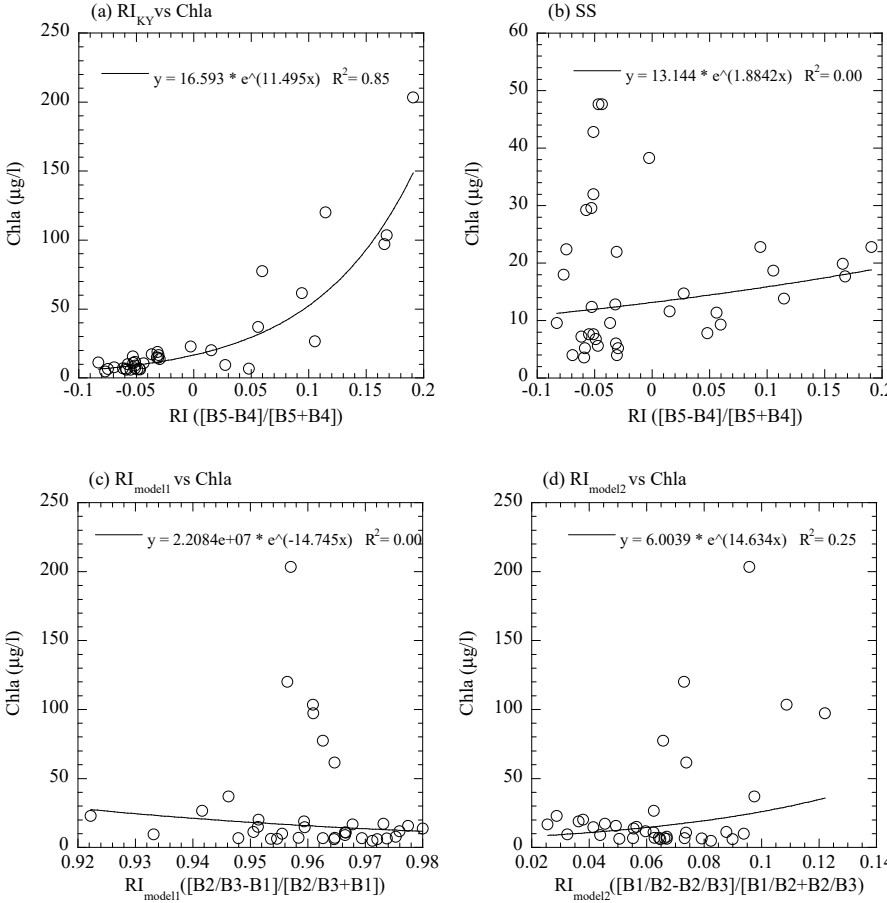

**Figure 4.** Relationship between tide index (*RI*) model combining two bands at 1 nm and water quality parameter. (**a**) $RI_{KY}$ vs chlorophyll a (Chla), (**b**) $RI_{KY}$ vs suspended substances (SS), (**c**) $RI_{model1}$ vs Chla, (**d**) $RI_{model2}$ vs Chla.

### 3.2. Validation of the Red Tide Model Using Sentinel-2 MSI

We validated the effectiveness of red tide monitoring by $RI_{KY}$ using field data in Section 3.1. Here, we validate the $RI_{KY}$ model using Sentinel-2 MSI data. Figure 5 shows the relationship of digital count (DC) with the reference date in MSI band 4 and band 5. Additionally, Table 4 shows the conversion equations and $R^2$ values for each day in each band. From this we see that, although the slope and offset slightly differ depending on the date, high correlation coefficients ($R^2 > 0.94$) were obtained in each case. The regression based on four points is weak in the statistical sense. In this case, only four suitable reference targets could be selected. Figure 6 shows the relationship between the $RI_{KY}$ model using MSI data and Chla before and after relative atmospheric correction. As a result, correlation was greatly improved from 0.19 to 0.93 after atmospheric correction. Moreover, this shape matches very well with the case using field reflectance data shown in Figure 4.

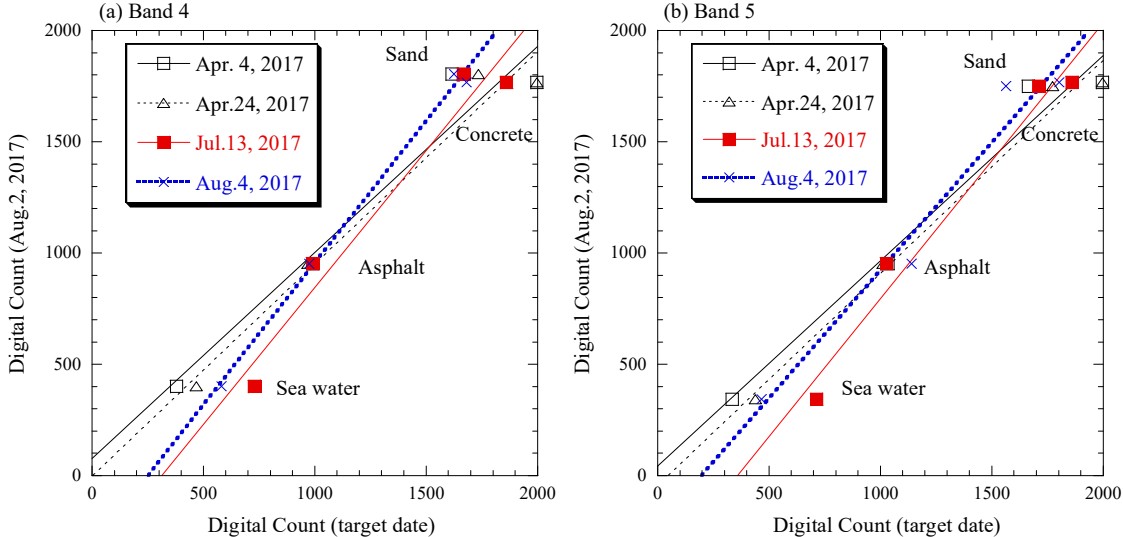

**Figure 5.** Relationship of digital count (DC) between target date and reference date (2 August 2017). (**a**) A case of Band 4, (**b**) A case of Band 5.

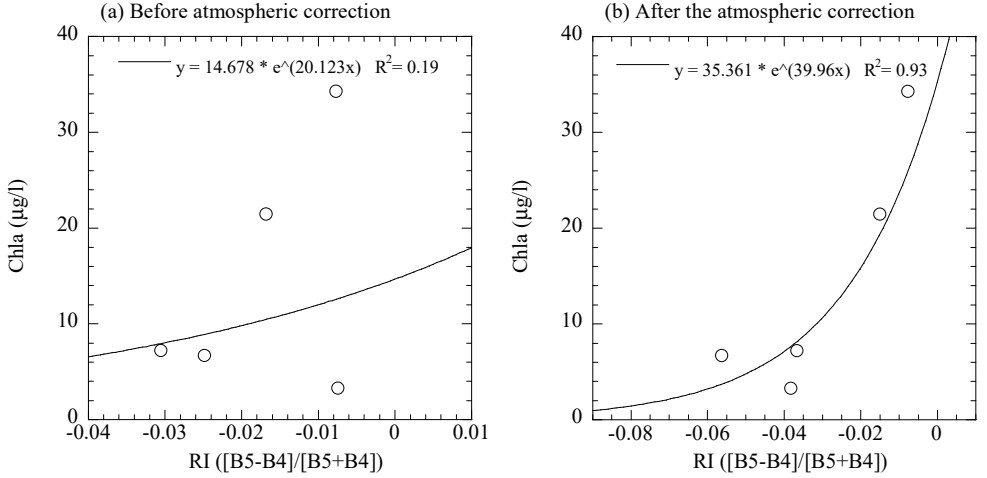

**Figure 6.** Relationship between tide index (*RI*) model 3 using Sentinel-2 MSI data and measured Chla (**a**) before atmospheric correction and (**b**) after atmospheric correction.

**Table 4.** Conversion equations and $R^2$ for relative atmospheric correction from digital count (DC) of Sentinel-2A MSI band 4 and band 5 on 2 August 2017 as a reference date.

| Band | Target Date | Conversion Equation | $R^2$ |
|---|---|---|---|
| B4 | 4 April | $y = 0.926x + 77.193$ | 0.94 |
| | 24 April | $y = 0.9521x + 0.8887$ | 0.97 |
| | 24 May | $y = 1.2312x - 384.31$ | 0.95 |
| | 13 July | $y = 1.2759x - 319.79$ | 0.98 |
| B5 | 4 April | $y = 0.926x + 77.193$ | 0.97 |
| | 24 April | $y = 0.9521x + 0.8887$ | 0.99 |
| | 24 May | $y = 1.2312x - 384.31$ | 0.97 |
| | 13 July | $y = 1.2759x - 319.79$ | 1.00 |

*3.3. Red Tide Distribution Characteristics Using Sentinel-2 MSI*

Figure 7 (left) shows Chla distribution and Figure 7 (right) shows red tide distribution in Lake Koyama-ike in 2017 using the $RI_{KY}$ model, the effectiveness of which was verified in Sections 3.1 and 3.2. Figure 7 (right) is the red tide distribution when the Chla threshold is set to 20 μg/L. There is almost no occurrence of red tides in the entire lake in April, and occurence is sparse in May and July on the southwest portion of the lake. It can also be seen that the red tide was distributed throughout the western part of the lake in August. We calculated the red tide ratio (area) to be 0%, 2.6% (0.2 km$^2$), 0.9% (0.1 km$^2$), and 73% (5.0km$^2$) of Lake Koyama-ike in April, May, July, and August, respectively. The red tide outbreak areas of May and July were all at the mouth of the inflowing river, whereas the red tides on 2 August were distributed as overhanging the right bank of Ao-shima Island in the southern part of the lake center. Such a distribution shows good agreement with the red tide distribution area as shown in Figure 8, which we confirmed using disaster prevention helicopter and field surveys in September 2017.

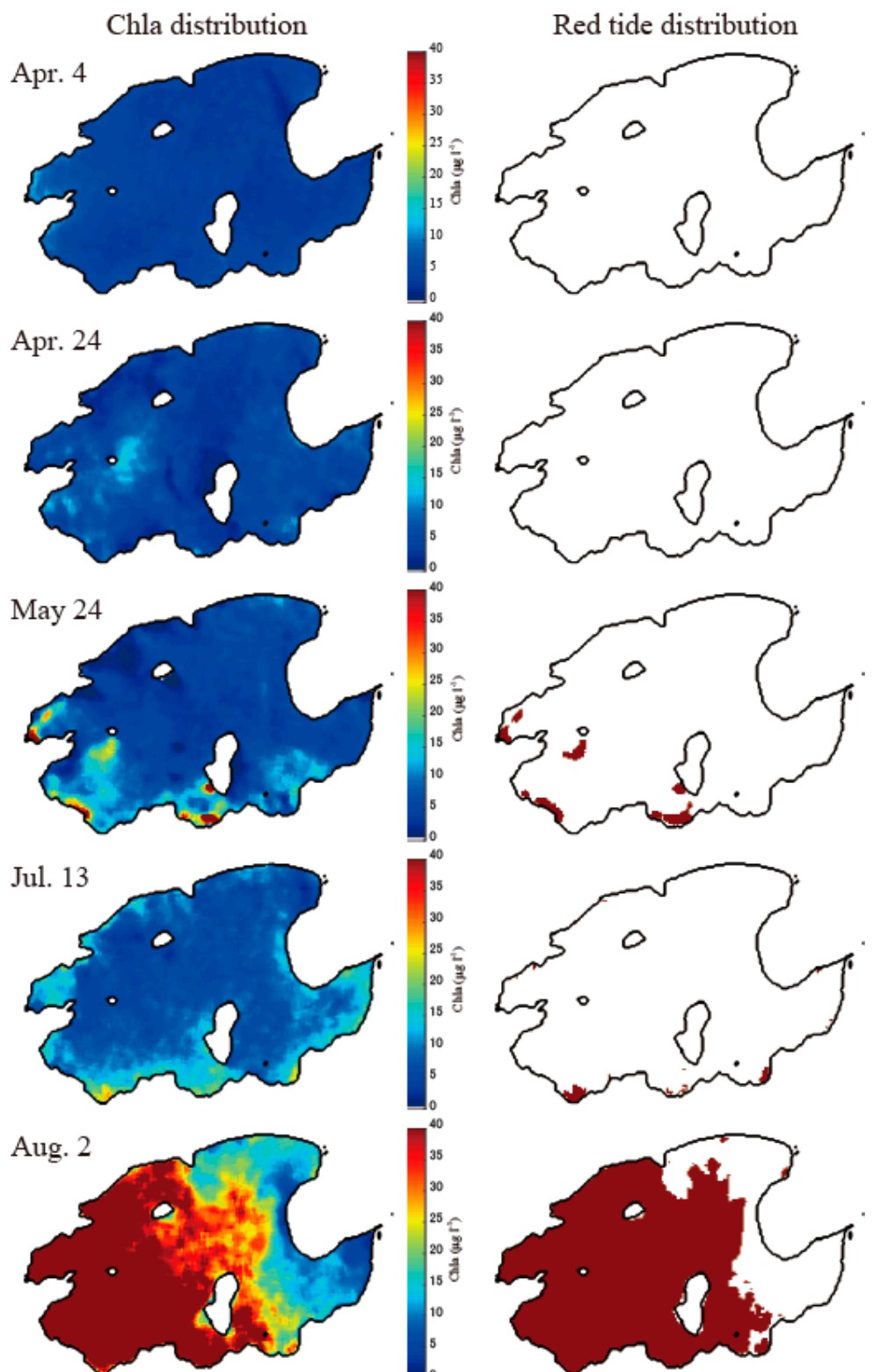

**Figure 7.** Chla and red tide bloom distribution maps in Lake Koyama-ike, 2017 by the proposal model ($RI_{KY}$) using Sentinel-2 data. Left figure shows Chla distribution and right figure shows red tide distribution.

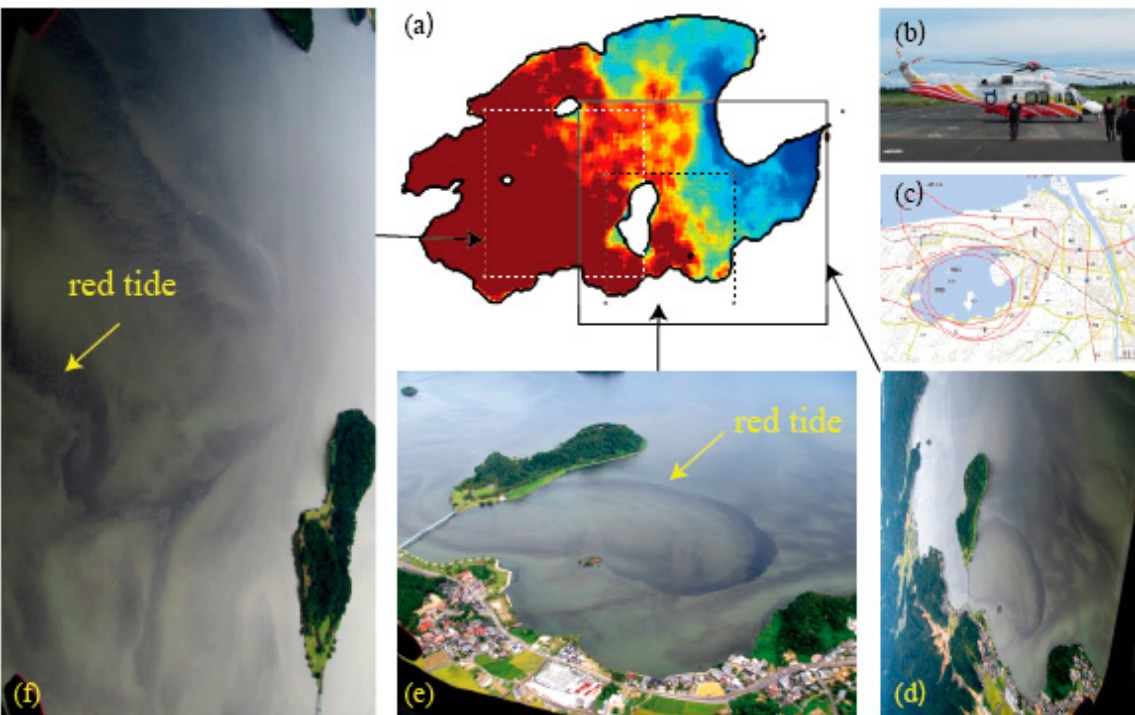

**Figure 8.** Comparison between chlorophyll A map derived from Sentinel-2 data on 2 August and an aerial photo taken from a disaster prevention helicopter on 4 September 2017. (**a**) Chla map from Sentinel-2 on 2 August, (**b**) disaster prevention helicopter used, (**c**) flight route (red line), and (**d**–**f**) aerial photos.

## 4. Discussion

The objective of this study is to propose and validate a red tide monitoring method in Lake Koyama-ike using Sentinel-2 MSI data. For the red tide model, we proposed a normalization model of Equation (6) based on a red-near infrared two-wavelength model that previous studies showed effective for high Chla estimation in coastal areas.

To achieve this objective, we first acquired spectral reflectance/Chla data in the field and verified the proposed model's validity. Next, we validated the proposed model using Sentinel-2 MSI data and automatically observed Chla data. Finally, we applied the proposed model to Sentinel-2 data, created red tide distribution maps for five periods from April to August 2017, and estimated the features and area proportions of red tide distributions.

Unlike MODIS and SeaWiFS, which estimate Chla using wavelengths near 400–550 nm, Sentinel-2 uses wavelengths of 665 nm (Band 4) and 705 nm (Band 5), which are effective for coastal Chla estimation. Success has been achieved in Chla estimation in various places [9,10]. In contrast, spectral reflectance characteristics of lakes have characteristic minimum and maximum values near 665 nm and 705 nm, respectively (Figure 2), and only the combination of the reflectance of Sentinel-2 band 4 and band 5 have shown a high correlation with Chla (Figure 3). This suggests that the Sentinel-2 MSI is a suitable sensor for Chla estimation in Lake Koyama-ike. Additionally, although we did not test it in this work for the sake of simplicity, it may be better to consider a 3-band algorithm [24,25] that is effective for Chla estimation of more turbid waters.

Chla estimation with Sentinel-2 MSI data requires complex atmospheric corrections using the Sen2cor algorithm [17]. However, using relative atmospheric correction [20,21], which is a simple atmospheric correction method, our proposed model can obtain high correlation results with local measurements regardless of season (Figure 5, Table 4, Figure 6). On the other hand, this relative atmospheric correction method was realized because there were relatively large and stable ground reference points such as open ocean, an airport, and a sandy beach near the lake basin. In general,

without such stable ground targets, atmospheric correction using the Sen2cor algorithm or the dark pixel method [18,19] will be necessary.

Phytoplankton species have been changed by increased salinity in Lake Koyama-ike after becoming brackish in 2013 [1]. As a result, problem in this area changed from blue-green alage to red tide; however, it was difficult to grasp red tide distribution and fluctuation because spatio-temporal changes were intense. On the other hand, Chla and red tide distributions estimated using Sentinel-2 data show that red tides did not occur in spring or early summer and then spread rapidly throughout the lake in late summer (August) of 2017 (Figure 7). Additionally, signs of red tides seen in the southeastern part of the lake in May and July may be due to nutrient runoff from inflowing rivers [26,27]. In particular, we identified a large-scale red tide in the southern part of Qingdao located in the central part of the lake from a disaster prevention helicopter on 4 September 2017, as shown in Figure 8, and its distribution is very similar to those from the model. These facts suggest that red tides may have continued for at least one month in the summer of 2017 in the western part of Lake Koyama-ike. In addition, this simple red tide mapping method can be used to manage red tides in relatively small lakes and red tide area can be quickly estimated because each pixel is 20m square. Thus, we found Sentinel-2 to be a very powerful tool for continuous monitoring of red tides in Lake Koyama-ike.

## 5. Conclusions

We proposed and validated a simple method for monitoring red tides in Lake Koyama-ike using field spectral reflectance/Chla measurements and the Sentinel-2/Chla data. As a result, the following matters became clear.

1. High correlation was obtained between the "$RI_{KY}$" using MSI band 4 and band 5 (proposed model) and Chla measurements from field spectral reflectance. Additionally, the relationship between the two matches better with the exponential regression model than with the linear regression model.
2. Chla estimation from Sentinel-2 data showed high correlation between the proposed model after relative atmospheric correction and Chla measurements.
3. Chla and red tide distribution characteristics estimated from Sentinel-2 data hardly appeared from April to July in 2017 and spread rapidly throughout the lake (more than 70%) in August.

From the above results, it is possible to easily create Chla and red tide maps through relative atmospheric correction automatically using Sentinel-2 data. In the future, we will study methods such as thin cloud and cloud shadow removal, correspondence of outliers, among other possibilities toward completely automatic red tide distribution processing. In addition, we would also like to examine a three-wavelength algorithm and construct a red tide detection method that can be used in brackish lakes along the coast of Japan.

**Author Contributions:** This study was designed by Y.S. and S.O. The field survey and data analysis were conducted by Y.S., A.M. (Akihiro Maeda), A.M. (Akihiro Mori), S.O. and A.I.

**Funding:** This research was funded by JSPS KAKENHI Grant Numbers, 17H04625, 19H04292.

**Conflicts of Interest:** The authors declare no conflict of interest.

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
