# Peer review of "A Simple Red Tide Monitoring Method using Sentinel-2 Data for Sustainable Management of Brackish Lake Koyama-ike, Japan"

_water, doi:10.3390/w11051044_

Round 1
Reviewer 1 Report
The article is a very important effort to provide information in short term on the variation in the lake water quality conditions, specially Chla, in several dates along the years, in dates with Red Tide development and other more usual conditions.
Is very valuable the use of remote sensors to compare methods and results in similar conditions with red tide and normal quality conditions of lake water, expanding the estimation to the whole water image.
Specially interesting is the selection of the sensor Sentinel 2 A/B, an excelent instrument for the estimation of Chla in the water (sensor bands are very useful to that purpose).
Must clarify some explanations, correct some errata, and other details to improve the understanding of the applied criteria (see the revised text with indication of some points to correct…) .
In general terms is OK the development of tools to estimate the aquatic parameters using the satellite sensor reflectances, and after the analysis of results and his correspondence with the in situ data.

Author Response
Thank you very much for receiving informative peer review comments. I was very impressed by your positive comments. All the red markers that you pointed out have been corrected as follows.
Comment 1. L81: tgate
Answer 1. As you pointed out, we corrected it to "gate" (L81).
Comment 2. L89: Alexandrium
Answer 2. This notation is not wrong, so we left it as it is. (L89).
Comment 3. L175: amosperic
Answer 3. As you pointed out, we corrected it to "atmospheric" (L179).
Comment 4. L176: Sentinael
Answer 4. As you pointed out, we corrected it to "Sentinel" (L180).
Comment 5. L300: alagae
Answer 5. As you pointed out, we corrected it to "algae" (L311).

Reviewer 2 Report
This manuscript proposed a index to detect red tide using Sentinel-2 imagery for the brackish lake Koyama-ike in Japan. Basically, this manuscript is clear-written and interesting. However, there are still some concerns as listed below.
1. Line 91-93. “Spectral reflectance/Chla and suspended substance (SS) data at points 6-15 in the lake (shown in Figure 1) were collected…” Revise this sentence. I guess you tried to say that the campaigns were conducted on the following dates and there are 6 to 15 measurements were collected during each campaign.
2. Line 81, “tgate” check spelling.
3. The dates for the campaigns are confusing. In Table 1, you listed the dates for the campaigns. Here, you have Aug. 22, 2018. Aug. 22 2018 also appeared in Line 93. But in Line 109, here comes the Aug. 2 (not sure is year 2017 or 2018). After Line 93, it looks like all the data are acquired before 2017. So, I’m confused did you have data in 2018?
4. Line 173-174. It is good that you have mentioned the difference between the spatial resolution between B5 and B4. You should also mention by which method you upscale the B4 to 20 m. For example, does the average method used?
5. Line 211-212. “ when combining the two bands at 1nm intervals in the 400-900nm range”. Please explain how you combine the two bands at 1nm intervals. I guess you mean that you used the reflectance measured by the spectroradiometer (MS 720) and the corresponding Chla measurements. The x-axis in Figure 3 is the first variable (Rrs(705) = the Band 2 in Figure 3) in equation (6) while the y-axis is the second variable (Rrs(665) = Band 1 in Figure 3) in equation (6). I suggest to explain more clear in the text and maybe add an equation (7) RIxy = Rrs(Band 2) - Rrs(Band 1) / Rrs(Band 2) + Rrs(Band 1).
6. Figure 2 caption. Make sure the date is correct. “non-red tide conditions using 6 data points on September 4, 2017.? According to Table 1, Sep. 4 2017 are in red tide bloom condition.
7. Figure 5, why use digital count to conduct regression? For more discussion about atmospheric correction, I will suggest to have a look at the following papers. You can find similar topic in [1] and detail review of atmospheric correction in [2-4].
8. Table 4, the regression based on four points is weak in statistical sense. Please add discussion in the text.
9. Figure 6, why only five points used here? This will also lead me to another question. Does the points in Figure 4 atmospheric corrected or not?
[1] Su, Y.F., Liou, J.J., Hou, J.C., Hung, W.C., Hsu, S.M., Lien, Y.T., Su, M.D., Cheng, K.S., Wang, Y.F., 2008, A multivariate model for coastal water quality mapping using satellite remote sensing images, Sensors, 8, 6321-6339.
[2] Cheng, K.S., Su, Y.F., Yeh, H.C., Chang, J.H., Hung, W.C., 2012, A path radiance estimation algorithm using reflectance measurements in radiometric control areas, International Journal of Remote Sensing, 33:5, 1543-1566.
[3] Schott, J. R., 1997, Remote Sensing – The image chain approach (New York: Oxford University Press).
[4 Switzer P. Kowalik, W., and Lyon, R.J.P., 1981, Estimation of atmospheric path-radiance by the covariance matrix method, Photogrammetric Enigneering and Remote Sesning, 47, 1469-1476.
Author Response
Thank you very much for your useful comments. We tried to modify as much as possible for your comment.
Comment 1. Line 91-93. “Spectral reflectance/Chla and suspended substance (SS) data at points 6-15 in the lake (shown in Figure 1) were collected…” Revise this sentence. I guess you tried to say that the campaigns were conducted on the following dates and there are 6 to 15 measurements were collected during each campaign.
Answer 1. As you pointed out, the expression of my text was not appropriate. Based on your advice, I have corrected it to the following sentences.
The field campaigns were conducted on the on July 19, September 25, 2002 (without red tide conditions), and on September 4, 2017 and August 22, 2018 (with red tide conditions). There are 6 to 15 measurements of spectral reflectance/Chla and suspended substance (SS) which were collected during each campaign as shown in Figure 1.
Comment 2. Line 81, “tgate” check spelling.
Answer 2. It is a mistake of "gate".
Comment 3. The dates for the campaigns are confusing. In Table 1, you listed the dates for the campaigns. Here, you have Aug. 22, 2018. Aug. 22 2018 also appeared in Line 93. But in Line 109, here comes the Aug. 2 (not sure is year 2017 or 2018). After Line 93, it looks like all the data are acquired before 2017. So, I’m confused did you have data in 2018?
Answer 3. I am sorry to be confused by our mistake. The sentences were corrected as follows
Thus it was judged that on September 4, 2017 and August 22, 2018, all stations and four out of six stations had red tide conditions, respectively. Additionally, during the red tide in 2017, we boarded a disaster prevention helicopter on September 4, 2017 to take photographs of the red tide from the air, and then collected surface water samples, which contained dinoflagellate Scrispsiella rotunda (the number of cells at the same point is 5.3 × 103 cells ml-1), a red tide plankton.
Comment 4. Line 173-174. It is good that you have mentioned the difference between the spatial resolution between B5 and B4. You should also mention by which method you upscale the B4 to 20 m. For example, does the average method used?
Answer 4. As you pointed out, since it was interpolated using the average method, it was corrected to the following sentences.
The spatial resolution of Band 4 has been scaled down to 20 m using average method to unify Band 4 and Band 5 resolutions used for Chla calculations.
Comment 5. Line 211-212. “ when combining the two bands at 1nm intervals in the 400-900nm range”. Please explain how you combine the two bands at 1nm intervals. I guess you mean that you used the reflectance measured by the spectroradiometer (MS 720) and the corresponding Chla measurements. The x-axis in Figure 3 is the first variable (Rrs(705) = the Band 2 in Figure 3) in equation (6) while the y-axis is the second variable (Rrs(665) = Band 1 in Figure 3) in equation (6). I suggest to explain more clear in the text and maybe add an equation (7) RIxy = Rrs(Band 2) - Rrs(Band 1) / Rrs(Band 2) + Rrs(Band 1).
Answer 5. As you pointed out, it was a confusing expression, so we added equation (7) and its descriptive text according to the advice.
Next, we investigated whether the wavelengths of MSI Band 4 and Band 5 for RIKY in Eq. (6) are appropriate. Figure 3 shows a matrix of correlation coefficients (R2) between Chla and the following RIxy when combining the two bands at 1 nm intervals in the 400–900 nm range to investigate suitability.
(7
Comment 6. Figure 2 caption. Make sure the date is correct. “non-red tide conditions using 6 data points on September 4, 2017.? According to Table 1, Sep. 4 2017 are in red tide bloom condition.
Answer 6. As you pointed out, it was the opposite. Therefore, the caption in Figure 2 has been corrected to the following document. Also, it was unified with "without / with red tide" instead of "non-red tide / red tide".
Figure 2. Field spectral reflectance characteristic in Lake Koyama-ike, Japan from 2012 to 2018. (a) All data, (b) comparison of average spectral reflectance for without red tide conditions using 9 data points on September. 25, 2012 and with red tide conditions using 6 data points on September 4, 2017.
Comment 7. Figure 5, why use digital count to conduct regression? For more discussion about atmospheric correction, I will suggest to have a look at the following papers. You can find similar topic in [1] and detail review of atmospheric correction in [2-4].
Answer 7. We agree with your point. It is more desirable to use the Radiance base rather than the DC base as in the paper originally introduced by you. However, since this study aims at a very simple method, the DC-based method used in the original relative atmospheric correction papers (Lopex Garcia et al., 1990; Oguma and Yamagata, 1997) were adopted. Therefore, adding the article (Su et al., 2008; Cheng et al., 2012) you introduced, the sentence was corrected as follows.
The relative atmospheric correction is originally desirable based on radiance or reflectance as in the reference [22, 23]. However, since we aim at a simple method in this research, it is calculated by DN-based method [20, 21].
22. Su, Y.F.; Liou, J.J.; Hou, J.C.; Hung, W.C.; Hsu, S.M.; Lien, Y.T.; Su, M.D.; Cheng, K.S.; Wang, Y.F., A multivariate model for coastal water quality mapping using satellite remote sensing images, Sensors, 2008, 8, pp.6321-6339.
23. Cheng, K.S. ; Su, Y.F. ; Yeh, H.C. ; Chang, J.H. ; Hung, W.C., A path radiance estimation algorithm using reflectance measurements in radiometric control areas, International Journal of Remote Sensing, 2012, 33(5), pp.1543-1566.
Comment 8. Table 4, the regression based on four points is weak in statistical sense. Please add discussion in the text.
Answer 8. We agree with your point. Essentially, it is desirable that the atmospheric correction be done with as many reference points as you can point out. However, only four stable reference targets were obtained this time. Therefore, the following sentences were newly added.
The regression based on four points is weak in statistical sense. In this case, only 4 suitable reference targets could be selected around, however it is desirable to use as much data as possible.
Comment 9. Figure 6, why only five points used here? This will also lead me to another question. Does the points in Figure 4 atmospheric corrected or not?
Answer 9. It may be confusing, but the data in Figure 6 is validated with the five satellite data shown in Table 3 at one validation point shown in Figure 1. Therefore, Figure 6 has 5 points only. Also, Figure 4 is a result of using field spectral reflectance data, so it has nothing to do with atmospheric correction.
Round 2
Reviewer 2 Report
The revised manuscript looks better and only one minor suggestion.
1. Line 91, “The field campaigns were conducted on the on July 19…” Drop the “on” before July.
Author Response
Thank you very much for your useful comments. We tried to modify as much as possible for your comment.
Comment 1. The revised manuscript looks better and only one minor suggestion.
1. Line 91, “The field campaigns were conducted on the on July 19…” Drop the “on” before July.
Answer 1. As you pointed out, "on" was overlapped and deleted.
That’s all,
Thank you.